



**Calibrating adsorptive and reactive losses of monoterpenes and sesquiterpenes in**
**dynamic chambers using deuterated surrogates**
Jianqiang Zeng [1,2], Yanli Zhang [1,2,*], Haofan Ran[1,2], Weihua Pang [1,2], Hao Guo[1], Zhaobin Mu [1], Wei Song [1],
Xinming Wang [1,2,*]
[1] State Key Laboratory of Organic Geochemistry and Guangdong Key Laboratory of Environmental
Protection and Resources Utilization, Guangzhou Institute of Geochemistry, Chinese Academy of Sciences,
Guangzhou 510640, China
[2] College of Resources and Environment, University of Chinese Academy of Sciences, Beijing 100049, China
*Correspondence: Xinming Wang (wangxm@gig.ac.cn) and Yanli Zhang (zhang_yl86@gig.ac.cn)



**Abstract**
Accurately measuring the emissions of monoterpenes (MTs) and sesquiterpenes (SQTs) using dynamic
chambers requires careful consideration of their adsorptive and reactive losses, which are often overlooked
and difficult to assess in situ. This study evaluated the effectiveness of deuterated surrogates, α-pinene-d3
and β-caryophyllene-d2, in tracing these losses in a dynamic chamber system. Using standard gas mixtures
of 10 MTs and 10 SQTs, we characterized adsorptive losses across varying concentrations, temperatures, and
humidity levels, as well as reactive losses with ozone. Results indicated that adsorptive losses were
significantly influenced by concentration and temperature, with species-specific variations particularly under
low concentrations and low temperatures, while relative humidity had negligible impact. Reactive losses with
ozone exhibited substantial species-specific variability. Key MTs (α-pinene, β-pinene, 3-carene, limonene,
and 1,8-cineole) and SQTs (β-caryophyllene and α-humulene) demonstrated consistent adsorptive and
reactive behavior with their respective deuterated surrogates α-pinene-d3 and β-caryophyllene-d2, suggesting
that these surrogates are effective for correcting losses in in-situ emission measurements using dynamic
chambers. However, due to varied adsorptive and reactive losses, additional deuterated MTs and SQTs are
recommended, particularly selected according to their $O_3$ reactivities, to cover a broader range of MTs and
SQTs for loss correction. A strong correlation between adsorptive capacity and ozone reactivity was observed,
underscoring the need to carefully address losses of highly reactive MTs and SQTs during emission
measurements. This study also emphasizes that ozone-free circulating air should be used for accurately
measuring emissions of highly reactive SQTs, such as β-caryophyllene and α-humulene, especially when loss
correction methods are unavailable.





## 1 Introduction

Monoterpenes (MTs) and sesquiterpenes (SQTs) emitted from terrestrial vegetation account for about 18% of global biogenic volatile organic compound (BVOC) emissions, with even higher contributions in certain ecosystems (Guenther et al., 2012). These reactive organic gases can rapidly react with atmospheric oxidants such as ozone ($O_3$), hydroxyl radicals (OH), and nitrate radicals ($NO_3$) (Di Carlo et al., 2004; Jardine et al., 2011; Edwards et al., 2017; He et al., 2021), influencing atmospheric oxidation capacity and leading to the formation of organic aerosols, which, as significant components of fine particles, can scatter solar radiation and act as cloud condensation nuclei (Yli-Juuti et al., 2021), directly and indirectly affecting air quality and climate (Peñuelas and Staudt, 2010; Unger, 2014; Scott et al., 2017).

Despite their importance, current BVOC emission models carry large uncertainties, largely due to inaccurate emission factors (Guenther et al., 2006, 2012). Common BVOC emission measurement techniques, such as leaf cuvettes and dynamic branch chambers (Niinemets et al., 2011; Šimpraga et al., 2019), are prone to adsorptive and reactive losses, particularly for the less volatile MTs and SQTs (Ortega et al., 2008). Their adsorptive losses, which can vary with conditions like temperature and concentration, may lead to significant underestimation of emission rates (Niinemets et al., 2011; Zeng et al., 2022a). For example, Helmig et al. (2004) reported that, after a 5-hour equilibrium, only 80% of SQTs and 60% of oxygenated SQTs were recovered in a leaf cuvette, requiring 5-10 hours to purge to background levels. Additionally, MTs and SQTs have short atmospheric lifetimes (from minutes to hours) due to their high reactivity with OH and $O_3$ (Atkinson and Arey, 2003). Field studies have shown significant in-canopy losses of SQTs (up to 61%) due to $O_3$ in the Amazonian rainforest (Jardine et al., 2011). These findings highlight the risk of underestimating the emissions of reactive terpenes using existing methods, especially in dynamic chambers where adsorptive and reactive losses remain a challenge (Niinemets et al., 2011).

Dynamic chambers, commonly made from chemically inert materials like Teflon or Tedlar, minimize but do not eliminate these losses (Niinemets et al., 2011; Materic et al., 2015). Previous studies have reported that adsorptive losses of MTs and SQTs in such chambers can exceed 30%, even with residence times of the chamber under one minute (Zeng et al., 2022a). Field studies also revealed that adsorptive losses in Tedlar bags reached 20-30% for MTs and ranged 10-80% for SQTs (Ortega et al., 2008). Temperature plays a critical role, with more substantial adsorptive losses at lower temperatures (Schaub et al., 2010). The adsorbed compounds may re-emit at higher temperatures, leading to potential overestimation (Ruuskanen et al., 2005; Schaub et al., 2010). Meanwhile, the presence of $O_3$ complicates measurements, as it both influences BVOC emissions from plants and reacts with MTs and SQTs (Feng et al., 2019; Zeng et al., 2022b). Some studies have used $O_3$-free circulating air to mitigate the reactive losses (Helmig et al., 2007; Ortega et al., 2008; Hellén et al., 2021), but this may neglect the impact of $O_3$ on BVOC emissions. In contrast, the use of ambient air can reflect real-world conditions (Kuhn et al., 2002; Bourtsoukidis et al., 2012; Zeng et al., 2022b), but the reactive loss existed. Despite reducing residence time by increasing flow rate of circulating air can lower





the reactive losses (Zeng et al., 2022a), it may still not be suitable for highly reactive MTs and SQTs
(Pollmann et al., 2005; Niinemets et al., 2011).
Addressing these issues is crucial for improving BVOC emission measurements. While internal surrogates
like aromatic compounds have been used to assess adsorptive losses, their suitability as surrogates for MTs
and SQTs remains debated (Helmig et al., 2007; Ortega et al., 2008). More importantly, the reactivity of these
compounds with $O_3$ in dynamic chambers has not been thoroughly evaluated (Pollmann et al., 2005; Helin et
al., 2020; Zeng et al., 2022a), and this reactive loss for individual MT and SQT species cannot be assessed
by using surrogates like aromatic compounds with quite different reactivity with $O_3$.
This study investigates the potential of using deuterated α-pinene-d3 and β-caryophyllene-d2 to quantify the
adsorptive and reactive losses of individual MTs and SQTs in dynamic chambers, taking advantage of the
almost identical adsorption and reaction behaviors of these deuterated compounds as their counterpart MTs
and SQTs. We evaluated these losses of α-pinene-d3 and β-caryophyllene-d2, along with target MTs and
SQTs, under varying concentration, temperature, and humidity, and analyzed the gas-phase reactive losses
with $O_3$ at different concentrations.
**2 Materials and methods**
**2.1 Lab evaluation**
A cylindrical dynamic chamber, which is made of polymethyl methacrylate with inner surface coated with
fluorinated ethylene propylene (FEP) Teflon film (FEP 100, Type 200A, DuPont, Witton, USA), was used
for lab evaluation. Detailed information about the design and initial characterization of this dynamic chamber
was described in our previous study (Zeng et al., 2022a). Here, we used standard gas mixtures of α-pinene-
d3, β-caryophyllene-d2, ten species of MTs, and ten species of SQTs to simulate their emissions from
enclosed leaves in the chamber. Detailed description of the preparation of the standard gas mixtures can be
found in Zeng et al. (2022a), and the information of standards were provided in Table S1. As shown in Fig.
S1, when the chamber reached a steady state offline samples were collected simultaneously from inlet ($C_0$)
and outlet ($C_1$) by commercial adsorbent cartridges (Tenax TA/Carbograph 5TD, Markes International Ltd,
Bridgend, UK) connected onto a portable dual-channel sampler (ZC-QL, Zhejiang Hengda Instrumentation
Ltd., Zhejiang, China) at a rate of 200 mL min$^{-1}$ for 5 min. Therefore, the recovery was expressed as $C_1/C_0$
in percentage (Fig. S1). More detail descriptions of the experiment are provided in Text S1.
The effect of concentration on adsorptive losses was evaluated under four concentration levels, with total
mixing ratios of MTs and SQTs of 0.23 and 0.26, 0.37 and 0.47, 0.68 and 0.94, and 2.26 and 3.16 ppb,
respectively. These experiments were conducted in $O_3$-free air at a temperature of approximately 25 °C and
a relative humidity of about 50%. The temperature effect was tested at two MTs and SQTs concentration
levels: a low concentration of 0.22 and 0.28 ppb and a high concentration of 2.26 and 3.16 ppb. In these
experiments, the relative humidity was about 50% but varied slightly with temperature, while the circulating
air was $O_3$-free. For the low-concentration experiments, the temperature ranged from 15 °C to 40 °C in 5 °C
increments, whereas the high-concentration experiments started at 10 °C and ended at 45 °C, also in 5 °C
increments. Humidity effect was tested under both low and high concentration levels, similar to the
temperature experiments. These humidity experiments were conducted at low (5%), mid (50%), and high
(95%) relative humidity (RH) levels, with a constant temperature of about 25 °C and $O_3$-free circulating air.
Additionally, the ozone effect was assessed at six ozone mixing ratios of 0, 30, 45, 60, 100, and 200 ppb,
with temperature of about 25 °C and RH of approximately 50%.

**2.2 Lab analysis**

Cartridge samples were analyzed by an automatic thermal desorption system (TD-100, Markes International
Ltd, Bridgend, UK) coupled to a mode 7890 gas chromatography (GC) with a model 5975 mass selective
detector (MSD) (Agilent Technologies, Inc., CA, USA). The adsorbent cartridges were thermally desorbed
by the TD-100 at 280 °C for 10 minutes and then the desorbed analytes were transferred by pure helium into
a cryogenic trap (U-T11PGC-2S, Markes International Ltd, Bridgend, UK) at -10 °C. Then the trap was
rapidly heated to transfer the analytes to the GC/MSD system with an HP-5MS capillary column (30 m ×
0.25 mm × 0.25 μm, Agilent Technologies, Inc., CA, USA). The GC oven temperature was programmed to
be initially at 35 °C (held for 3 minutes), increased to 100 °C at 5 °C min$^{-1}$ and held for 1 minute, then to
120 °C at 10 °C min$^{-1}$ and held for 12 minutes, and then to 260 °C with a final hold time of 2 minutes. The
MSD was operated simultaneously under scan mode and selected ion monitoring mode with electron
impacting ionization at 70 eV. Target compounds were identified by comparing their retention times with
authentic standards (Table S1), and were quantified with the standard calibration curves. More detailed
descriptions of the identification and quantification can refer to our previous studies (Zeng et al., 2022b; Zeng
et al., 2023; Zeng et al., 2024). Method detection limits (MDLs) were determined by seven parallel analyses
of the lowest concentration of calibration standards. Based on a sample volume of 1 L, the MDLs varied from
8 to 92 ng m$^{-3}$ for MTs and from 2 to 92 ng m$^{-3}$ for SQTs, and the measurement precision ranged from 2.3 to
4.8 % for MTs and from 1.1 to 3.4 % for SQTs (Table S2).

**3 Results and discussion**

**3.1 Concentration effect**

The adsorptive behaviors of different MTs and SQTs varied significantly across concentration levels (Fig. 1).
At low concentrations, recoveries ranged from 59% for γ-terpinene to 81% for α-pinene among MTs and
from 40% for α-humulene to 71% for longicyclene among SQTs (Fig. 1a). In contrast, under high
concentration levels, recoveries were more consistent, ranging from 75% for β-myrcene to 87% for β-pinene
among MTs and from 74% for β-chamigrene to 84% for α-longipinene for SQTs (Fig. 1d). Overall, recoveries





were higher at elevated concentrations (Figs. S3 and S4), with average recoveries of 68±9%, 75±7%, 78±5%,
and 84±2% for all MTs, and 58±8%, 63±5%, 71±4%, and 82±1% for all SQTs across four concentration
levels (Fig. S5).
Despite using chemically inert materials in chamber construction, significant adsorptive losses of MTs and
SQTs persist (Ortega et al., 2008; Niinemets et al., 2011; Kolari et al., 2012; Zeng et al., 2022a). Our findings
corroborate previous studies, demonstrating that low concentrations lead to higher relative losses (Fig. 1). In
steady-state conditions, the amount of adsorbed MTs and SQTs is constant, implying that lower concentration
results in a greater proportion of total loss (Figs. S3 and S4). This suggests that low-emitting plants, especially
for SQTs, may experience substantial adsorptive losses.
We observed species-specific recovery differences at low concentrations (Fig. 1a), likely due to their
physicochemical properties (Niinemets et al., 2004, 2011). Less reactive MTs like α-pinene-d3, α-pinene, β-
pinene, and 1,8-cineole exhibited stable recoveries regardless of concentration changes. In contrast, more
reactive MTs, such as β-myrcene, α-phellandrene, γ-terpinene, and terpinolene, showed significant recovery
variations (Fig. S3). This pattern also applies to SQTs, with highly reactive compounds, like β-caryophyllene,
β-caryophyllene-d2, and α-humulene, being more affected than others (Fig. S4). This compound-specific
adsorptive losses indicate that α-pinene-d3 and β-caryophyllene-d2 cannot serve as surrogates for all MTs
and SQTs, respectively.
**3.2 Temperature effect**
Overall recoveries of MTs and SQTs were higher at elevated temperatures, with a smaller absolute effect
observed at high concentrations compared to low concentrations (Figs. 2 and S6). At the high concentration,
mean recoveries increased from 80±4% to 91±1% for MTs and from 73±3% to 89±3% for SQTs as
temperature rose from 10 to 45 °C (Figs. S7 and S8). At the low concentration, temperature influenced
recoveries more significantly (Fig. 2), with mean values rising from 70±12% to 85±4% for MTs and from
50±21% to 77±6% for SQTs as temperature increased from 15°C to 40°C. At low concentration and low
temperatures (e.g., 15 °C), the standard deviation of recoveries was greater, such as 12% for MTs and 21%
for SQTs, indicating variability among species (Fig. 2). However, at low concentration but high temperatures
(e.g., 40°C), recoveries became more consistent, with reduced standard deviation of 4% for MTs and 6% for
SQTs (Fig. 2).
The temperature effect on recovery was compound-specific. Among MTs, β-myrcene, α-phellandrene, γ-
terpinene, terpinolene, and linalool showed greater sensitivity (Fig. 2a), while β-caryophyllene, β-
caryophyllene-d2, and α-humulene showed greater sensitivity among SQTs (Fig. 2b). We categorized MTs
and SQTs into two groups based on their adsorptive behaviors (Fig. 3). MTs in group 1 included α-pinene-
d3, α-pinene, β-pinene, 3-carene, limonene, and 1,8-cineole, while MTs in group 2 consisted of β-myrcene,



α-phellandrene, γ-terpinene, terpinolene, and linalool. SQTs in group 1 comprised α-longipinene,
longicyclene, α-copaene, α-gurjunene, thujopsene, aromadendrene, alloaromadendrene, and β-chamigrene,
whereas SQTs in group 2 included β-caryophyllene, β-caryophyllene-d2, and α-humulene. Under the high
concentration, temperature changes minimally affected recoveries, with slopes of $0.37\pm0.07\%$ °C$^{-1}$ for all
MTs and $0.43\pm0.08\%$ °C$^{-1}$ for all SQTs (Fig. 3). However, under the low concentration, sensitivity varied
significantly between groups, with slopes of $0.27\pm0.06\%$ °C$^{-1}$ for MTs and $0.53\pm0.05\%$ °C$^{-1}$ for SQTs in
group 1 much lower than those $1.12\pm0.12\%$ °C$^{-1}$ for MTs and $2.51\pm0.19\%$ °C$^{-1}$ for SQTs in group 2 (Fig. 3).
Our results demonstrate that adsorptive losses of MTs and SQTs are influenced by both concentration and
temperature. Elevated temperatures facilitate evaporation and diffusion, reducing adsorptive losses
(Niinemets et al., 2004; Ortega and Helmig, 2008). Field measurements on *Picea pungens* have similarly
shown increased recoveries at high temperatures (Ortega et al., 2008). At low concentrations, the pronounced
sensitivity to temperature results in greater recovery differences among species particularly at lower
temperatures (Fig. 3), indicating that α-pinene-d3 and β-caryophyllene-d2 may not effectively calibrate
adsorptive losses across all MTs and SQTs. Since the inter-group adsorptive behaviors (e.g., MTs in group 1
vs. in group 2) differ significantly but the intra-group adsorptive behaviors for both MTs and SQTs remain
consistent (Fig. 3), suggesting that α-pinene-d3 and β-caryophyllene-d2 instead can be used to calibrate the
adsorptive losses for MTs in group 1 (α-pinene, β-pinene, 3-carene, limonene, and 1,8-cineole) and SQTs in
group 2 (β-caryophyllene and α-humulene), respectively.

### 3.3 Humidity effect

The impact of humidity on the adsorptive losses of MTs and SQTs was examined at three RH levels: 5%,
50%, and 95%. Overall recoveries were slightly higher at 95% RH compared to lower RH levels (Fig. 4). For
low-concentration experiments, mean recoveries of MTs and SQTs were $72\pm9\%$ and $64\pm20\%$ at 5% RH,
$72\pm10\%$ and $62\pm13\%$ at 50% RH, and $80\pm6\%$ and $70\pm15\%$ at 95% RH, respectively (Fig. 4a), while for
high-concentration experiments, they were $77\pm2\%$ and $76\pm2\%$ at 5% RH, $79\pm5\%$ and $82\pm2\%$ at 50% RH ,
and $82\pm2\%$ and $82\pm2\%$ at 95%, respectively (Fig. 4b), with recoveries more consistent across RH levels.
The results suggests that humidity has a minor impact on adsorptive loss, aligning with previous findings
(Hohaus et al., 2016; Helin et al., 2020; Zeng et al., 2022a). The relationship between adsorptive losses and
humidity is complex, as water molecules can compete with MTs and SQTs for adsorption sites, potentially
reducing the adsorption of hydrophobic MTs and SQTs (Kolari et al., 2012; Zeng et al., 2022a). These results
indicate that the effect of humidity on the adsorptive loss of MTs and SQTs in dynamic chambers can be
considered negligible during field studies.

### 3.4 Ozone effect

The impact of O$_3$ on the reactive loss of MTs and SQTs was evaluated at six mixing ratio levels: 0, 30, 45,





60, 100, and 200 ppb. Recoveries of α-pinene-d3, α-pinene, β-pinene, 3-carene, and 1,8-cineole remained
stable across $O_3$ levels. However, recoveries of β-myrcene, limonene, and γ-terpinene declined slightly with
elevated $O_3$ levels, and more pronounced losses were observed for α-phellandrene and terpinolene, with
recoveries of 70±3% and 78±2% at an $O_3$ mixing ratio of 60 ppb, respectively (Fig. S9). For SQTs,
longicyclene, α-copaene, aromadendrene, and alloaromadendrene exhibited relatively stable recoveries,
while α-longipinene, α-gurjunene, thujopsene, and β-chamigrene suffered significant losses (Fig. 5). Notably,
the recoveries of β-caryophyllene, β-caryophyllene-d2, and α-humulene dropped sharply with increasing $O_3$
levels, with recoveries as low as 41±3%, 36±6%, and 30±4%, respectively, even at a low $O_3$ level of 30 ppb
(Fig. 5).
Given ozone's role in reactive losses, some studies have used $O_3$-free circulating air during field
measurements to mitigate these effects (Ortega et al., 2008; Helmig et al., 2013). However, it is also critical
to understand ozone's role in affecting emissions of MTs and SQTs (Bourtsoukidis et al., 2012; Feng et al.,
2019; Zhang et al., 2022). Since the measured emissions may not reflect real-world emissions if $O_3$ is cleaned,
some studies have used ambient air as circulating air during field measurements (Kuhn et al., 2002;
Bourtsoukidis et al., 2012; Zeng et al., 2022b). The relationship between reactive loss and $O_3$ concentration
can be modelled by:
$$C_t = C_0 \cdot exp \, (-k \cdot t \cdot [O_3]) \qquad (1)$$
where $C_0$ (molecule $cm^{-3}$) and $C_t$ (molecule $cm^{-3}$) are the initial and time $t$ concentrations in the chamber,
respectively; $k$ ($cm^3$ molecule$^{-1}$ s$^{-1}$) is the reaction rate constant; $t$ (s) is the reaction time of the compound in
the chamber; $[O_3]$ (molecule $cm^{-3}$) represents the concentration of $O_3$. In this study, when the $C_t$ reached to
equilibrium concentration ($C_s$) with a reaction time of the residence time, Eq. 1 can be rewritten as:
$$-ln \, (C_s/C_0) = k \cdot \frac{V}{F} \cdot [O_3] \qquad (2)$$
where $V$ (L) is the volume of the chamber and $F$ (L s$^{-1}$) is the flow rate of circulating air. As a result, through
the application of a linear regression between $-ln \, (C_s/C_0)$ and $[O_3]$, we can derive $k$ values for MTs and SQTs,
as illustrated in Fig. S10. However, $k$ values for some less reactive MTs and SQTs cannot be obtained this
way and Table 1 presents the $k$ values of some highly reactive species. In present study, the $k$ values of β-
myrcene, α-phellandrene, limonene, and terpinolene were 5.93±0.17, 28.15±1.11, 3.06±0.14, and 19.21±1.20
(×10$^{-16}$) $cm^3$ molecule$^{-1}$ s$^{-1}$, respectively, which are in agreement with those reported in literatures (Table 1).
For SQTs, the $k$ values were 20.17±0.98, 14.56±0.15, 115.33±3.83, 129.14±4.42, 8.64±0.16, 147.04±3.87,
and 15.14±0.16 (×10$^{-16}$) $cm^3$ molecule$^{-1}$ s$^{-1}$ for α-longipinene, α-gurjunene, β-caryophyllene, β-
caryophyllene-d2, thujopsene, α-humulene, and β-chamigrene, respectively. The $k$ values for the most
important β-caryophyllene and α-humulene were consistent with those reported previously (Table 1). The





good consistency of $k$ values with literatures proves that the assessment of reactive losses with $O_3$ in the
dynamic chamber used here is reliable.
The compound-specific reactive losses due to their varying reactivity with $O_3$ (Table 1) necessitates their
consideration during field measurements, particularly for highly reactive species like β-caryophyllene and
α-humulene. Reactive loss caused by $O_3$ can be reduced by shortening the residence time through increasing
the flow rate of circulating air. For the chamber in this study, with a flow rate of 9 L min$^{-1}$ for circulating air
(~1.5 minutes of residence time), the reactive losses of all tested MTs, except for α-phellandrene and
terpinolene, were less than 10%, even at a high $O_3$ mixing ratio of 100 ppb (Fig. S9). This suggests that the
effect of $O_3$ on the reactive losses of main MTs (e.g., α-pinene, β-pinene, limonene) can be disregarded during
field measurements. For SQTs, although the reactive losses for longicyclene, α-copaene, aromadendrene, and
alloaromadendrene were less than 10% at 100 ppb $O_3$, more than 60% of the critical species like β-
caryophyllene and α-humulene were consumed by $O_3$ even at a low mixing ratio of 30 ppb (Fig. 5). As shown
in Fig. S11, to limit the reactive loss of β-caryophyllene within 15%, the flow rate needs to exceed 42 L min$^{-}$
$^1$ while $O_3$ mixing ratio should be kept below 30 ppb. Although the detection of instrument is not a problem
even under this high flow rate (Zeng et al., 2022a), the requirement for an $O_3$ mixing ratio below 30 ppb
poses practical challenges, as this level is already lower than those observed in most background regions
worldwide (Lu et al., 2018; Wang et al., 2019). This suggests that using $O_3$-free circulating air may be more
suitable when measuring emissions of highly reactive SQTs in the field, as a short-term $O_3$ scavenge during
measurements could have negligible impact on the BVOC emissions as demonstrated by Niinemets et al.

249    (2011).

**3.5 Implications for field measurements**
As discussed above and illustrated in Fig. 6, the effects of concentration, temperature, and $O_3$ on the losses
of MTs and SQTs are compound-specific. The adsorptive and reactive losses in group 2 are generally greater
than those in group 1, reflecting their different physicochemical properties (Niinemets et al., 2011). Notably,
we observed a positive correlation between adsorptive capacity and $O_3$ reactivity (Fig. 7), suggesting that
compounds with higher $O_3$ reactivity experience greater temperature- and concentration-driven adsorptive
losses. These findings underscore the importance of accounting for both reactive and adsorptive losses in
accurately measuring emissions of highly reactive MTs and SQTs.
Furthermore, findings here demonstrate temperature as a crucial factor in influencing the adsorptive losses
of terpenes especially SQTs in dynamic chambers. For instance, assuming leaves with a dry mass of 10 g are
enclosed in the chamber, the lowest concentration evaluated here, equivalent to emission rates of 0.07 μg g$^{-1}$
h$^{-1}$ for MTs and 0.12 μg g$^{-1}$ h$^{-1}$ for SQTs, represents a low MT-emitting but high SQT-emitting plant (Duhl et
al., 2008; Ortega et al., 2008). Despite being a low MT emitter, at temperatures below 30 °C, the adsorptive
losses for MTs in group 1 are only about 20%, while those in group 2 exceed 30%. In contrast, adsorptive





losses for SQTs in groups 1 and 2 can be greater than 30% and 45%, respectively, even for a high SQT-
emitting plant. This suggests that in high-latitude regions with relatively cold weather, while adsorptive losses
of some key MTs, such as those in group 1, may be acceptable, losses can be significant for SQTs especially
the most important β-caryophyllene and α-humulene, thus needing loss corrections during measurements.
Moreover, in tropical and subtropical regions characterized by hot weather, high daytime temperatures and
strong sunlight can cause temperatures in dynamic chambers to exceed 35 °C, even surpassing 45 °C during
heat events (Zeng et al., 2022b; Doughty et al., 2023). Under these conditions, adsorptive losses can be
reduced to within 20% for most MTs and 25% for most SQTs, regardless of changes in concentration and
humidity. This indicates that adsorptive losses of MTs and SQTs may not pose a significant issue for dynamic
chambers used in hot regions; however, the reactive losses of highly reactive compounds must still be
carefully considered to ensure accurate measurement of their emission rates.
Although species-specific variabilities in adsorptive and reactive losses existed, we demonstrated that MTs
in group 1, including α-pinene, β-pinene, 3-carene, limonene, and 1,8-cineole, exhibit consistent adsorptive
behavior with α-pinene-d3, regardless of changes in concentration, temperature, or humidity, and the reactive
losses of these MTs are minimally affected by ambient-level $O_3$. This consistency makes α-pinene-d3 a
suitable surrogate. Similarly, β-caryophyllene-d2 aligns well with β-caryophyllene and α-humulene, and is a
suitable surrogate for these compounds. However, MTs in group 2 and SQTs in group 1 exhibit significant
differences in adsorptive and reactive behaviors compared to α-pinene-d3 and β-caryophyllene-d2,
respectively, and therefore cannot be calibrated using these two deuterated surrogates. Given the strong
correlation between adsorptive capacity and $O_3$ reactivity (Fig. 7), we recommend considering other
deuterated MTs and SQTs that share similar adsorptive and reactive behaviors with those in group 2 and
SQTs in group 1 as internal surrogates for loss correction.
**4 Conclusions**
In this study, lab evaluations identified significant species-specific variability in the adsorptive and reactive
losses of MTs and SQTs in dynamic chambers, influenced by their distinct physicochemical properties.
Adsorptive losses were notably influenced by concentration and temperature, with pronounced variability at
low concentrations and low temperatures. The reactive losses were also species-specific, with highly reactive
compounds like β-caryophyllene and α-humulene exhibiting significant losses. This emphasizes the
importance of using $O_3$-free circulating air during emission measurement to minimize these losses for highly
reactive SQTs.
This study supports the use of α-pinene-d3 and β-caryophyllene-d3 as effective surrogates for loss correction.
These surrogates showed consistent adsorptive and reactive behaviors with corresponding target compounds
(e.g., α-pinene, β-pinene, 3-carene, limonene, and 1,8-cineole for MTs, and β-caryophyllene and α-humulene
for SQTs), enabling more accurate calibration of losses during in-situ measurements. Additionally, the



298 observed correlation between adsorptive capacity and ozone reactivity indicates that other deuterated MTs

299 and SQTs could be employed to expand coverage for loss correction.

300 Our study demonstrates that adsorptive losses in dynamic chambers are highly temperature dependent. In

301 tropical and subtropical regions, higher temperatures potentially reduce adsorptive losses of MTs and SQTs

302 to acceptable levels. Conversely, low temperatures typical of high-latitude regions could lead to significant

303 adsorptive losses, particularly for SQTs, necessitating careful correction.

304 To ensure accurate emission measurements, it is crucial to account for both adsorptive and reactive losses of

305 MTs and SQTs, especially those with high reactivity. For highly reactive SQTs like β-caryophyllene and α-

306 humulene, $O_3$-free circulating air should be used to avoid substantial reactive losses. Furthermore, selecting

307 internal surrogates that closely match the adsorptive and reactive properties of target compounds is vital for

308 precise loss correction.

309

310 ***Data availability***

311 All data used in this study are provided in the manuscript and/or the supplement.

312

313 ***Supplement***

314 The supplement related to this article may be available online.

315

316 ***Author contributions***

317 JZ designed and carried out the experiments with the support of XW, YZ, WS, and HG. JZ, HR, WP, and ZM

318 analyzed the samples in the lab. JQZ analyzed the data and prepared the original manuscript. XMW and YLZ

319 revised the manuscript.

320

321 ***Competing interests***

322 The authors declare that they have no conflict of interest.

323



### Financial supports

This work was supported by the National Natural Science Foundation of China (Nos. 42022023 and 42321003), the National Key Research and Development Program (2022YFC3701103), the Youth Innovation Promotion Association, CAS (Y2021096), the Department of Science and Technology of Guangdong (Nos. 2020B1111360001, 2023B0303000007 and 2023B1212060049), and the Guangzhou Municipal Science and Technology Bureau (No. 202206010057).

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





**Figure captions**
Figure 1. Recoveries of MT and SQT species under four different concentration levels. Total MTs and SQTs
mixing ratios are 0.23 and 0.26 ppb (a), 0.37 and 0.47 ppb (b), 0.68 and 0.94 ppb (c), and 2.26 and 3.16 ppb
(d), respectively.
Figure 2. Recoveries of different MT (a) and SQT (b) species under different temperatures under the low
concentration level.
Figure 3. Variations of recoveries of MTs and SQTs along with temperature under low and high concentration
levels. MTs group 1 includes α-pinene-d3, α-pinene, β-pinene, 3-carene, limonene, and 1,8-cineole, while
MTs group 2 includes β-myrcene, α-phellandrene, γ-terpinene, terpinolene, and linalool. α-Longipinene,
longicyclene, α-copaene, α-gurjunene, thujopsene, aromadendrene, and alloaromadenrene belong to SQTs
group 1, while β-caryophyllene, β-caryophyllene-d2, and α-humulene belong to SQTs group 2.
Figure 4. Recoveries of different MT and SQT species under different relative humidity levels; (a) low
concentration level, (b) high concentration level.
Figure 5. Reactive losses of SQT species under different $O_3$ mixing ratios.
Figure 6. The relationships between loss ratio of different MTs and SQTs groups with concentration,
temperature, and ozone mixing ratio.
Figure 7. Relationship of ozone reactivity with the effects of concentration (a) and temperature (b); The conc.
effect represents the absolute change of adsorptive loss per ppb (|a|) by fitting the loss ratio and the total
mixing ratios of MTs or SQTs using $y = a \cdot \ln(x) + b$ as in Fig. 6; The temp. effect represents the absolute change
of adsorptive loss per degree Celsius (|a|) by fitting the loss ratio and the temperature using $y = a \cdot x + b$ as in
Fig. 6; Each point represents an individual MT or SQT species.





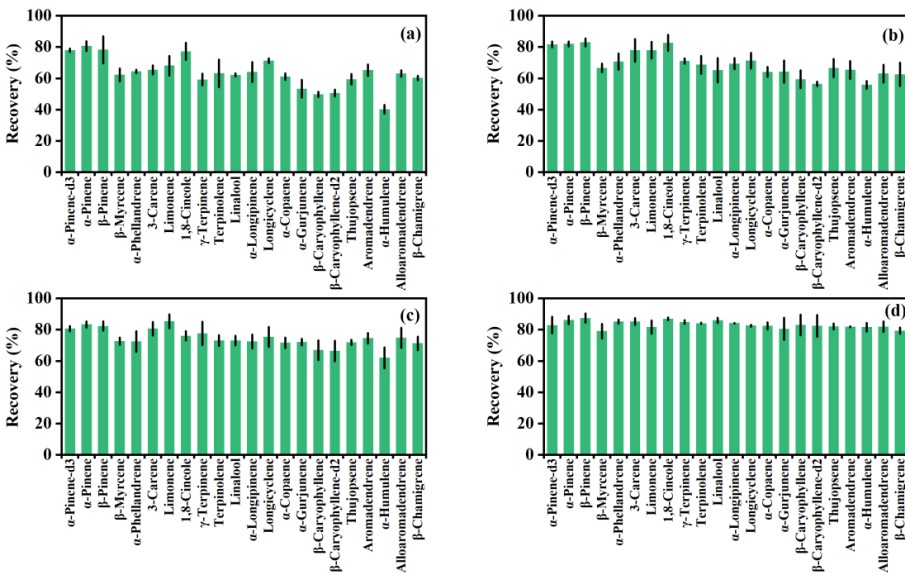

**Figure 1.** Recoveries of MT and SQT species under four different concentration levels. Total MTs and SQTs mixing ratios are 0.23 and 0.26 ppb (a), 0.37 and 0.47 ppb (b), 0.68 and 0.94 ppb (c), and 2.26 and 3.16 ppb (d), respectively.





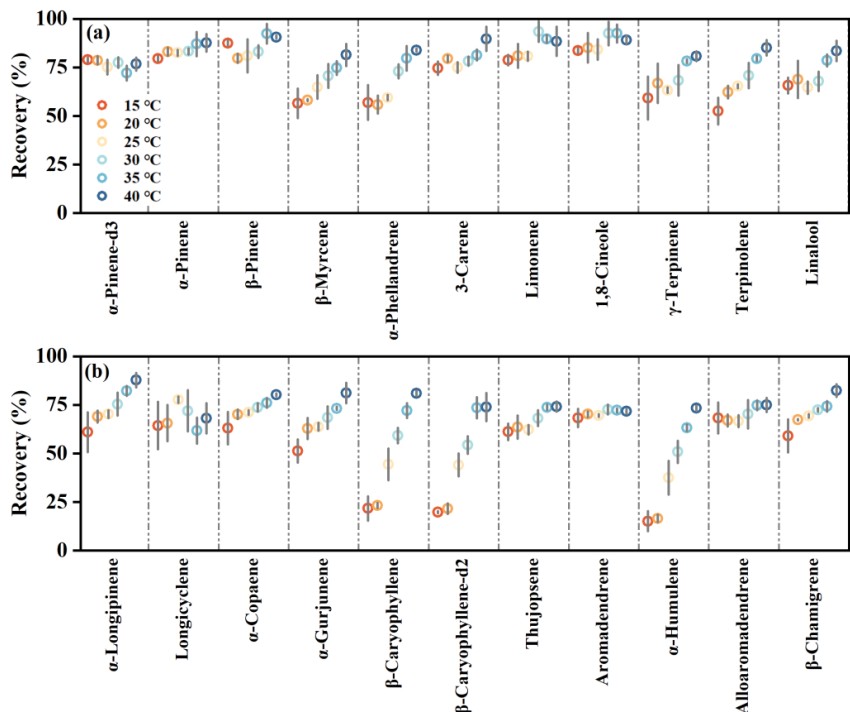

**Figure 2.** Recoveries of different MT (a) and SQT (b) species under different temperatures under the low concentration level.





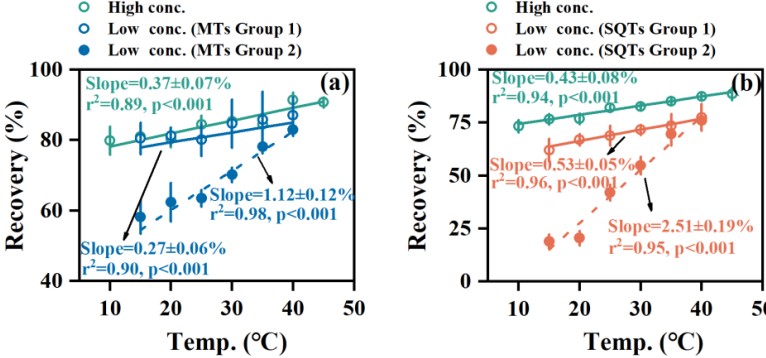

**Figure 3.** Variations of recoveries of MTs and SQTs along with temperature under low and high concentration levels. MTs group 1 includes α-pinene-d3, α-pinene, β-pinene, 3-carene, limonene, and 1,8-cineole, while MTs group 2 includes β-myrcene, α-phellandrene, γ-terpinene, terpinolene, and linalool. α-Longipinene, longicyclene, α-copaene, α-gurjunene, thujopsene, aromadendrene, and alloaromadendrene belong to SQTs group 1, while β-caryophyllene, β-caryophyllene-d2, and α-humulene belong to SQTs group 2.





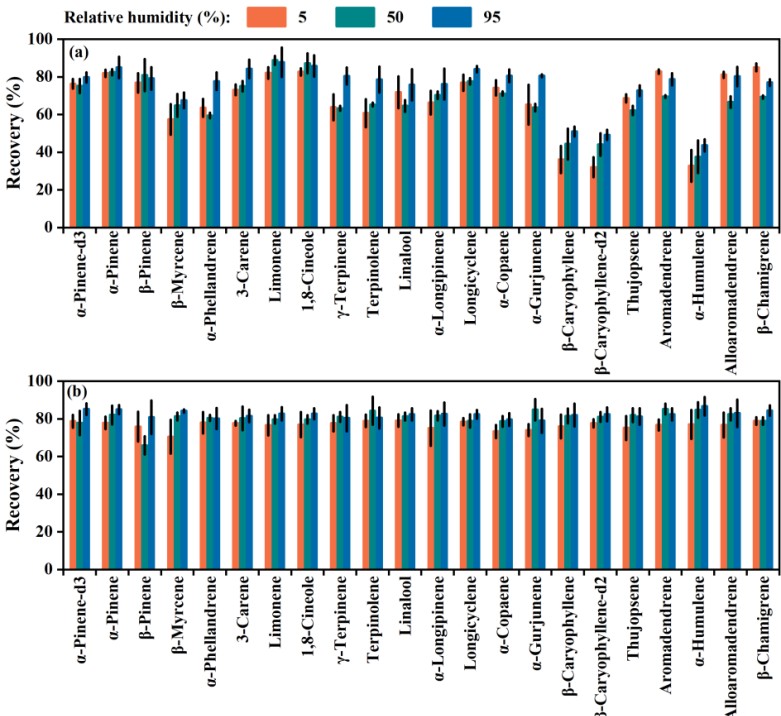


**Figure 4.** Recoveries of different MT and SQT species under different relative humidity levels; (a) low concentration level, (b) high concentration level.






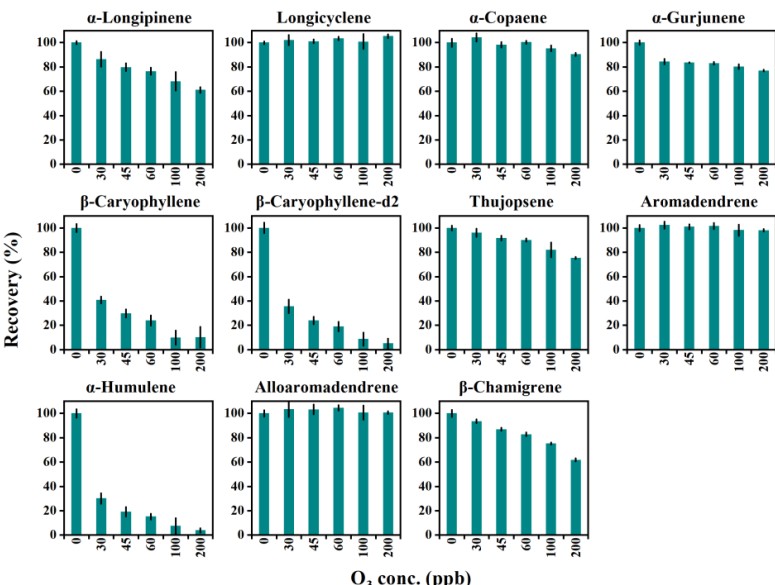


**Figure 5.** Reactive losses of SQT species under different O₃ mixing ratios.





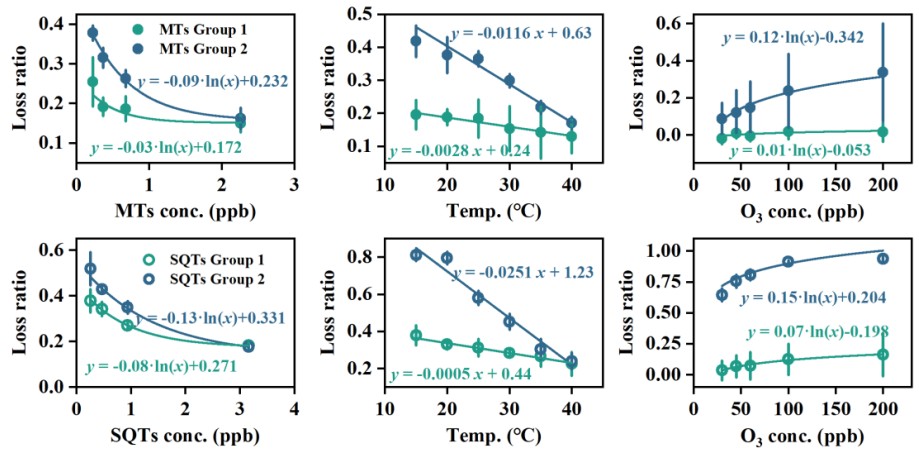


**Figure 6.** The relationships between loss ratio of different MTs and SQTs groups with concentration, temperature, and ozone mixing ratio.






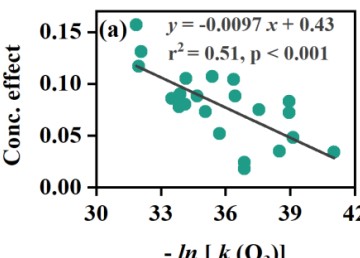
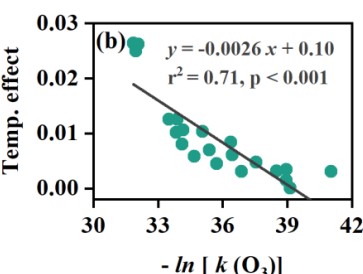


**Figure 7.** Relationship of ozone reactivity with the effects of concentration (a) and temperature (b); The conc.
effect represents the absolute change of adsorptive loss per ppb (|a|) by fitting the loss ratio and the total
mixing ratios of MTs or SQTs using $y = a \cdot \ln(x) + b$ as in Fig. 6; The temp. effect represents the absolute change
of adsorptive loss per degree Celsius (|a|) by fitting the loss ratio and the temperature using $y = a \cdot x + b$ as in
Fig. 6; Each point represents an individual MT or SQT species.






**Table 1.** The measured and literature reported rate constants ($k$, $\times 10^{-16}$ cm$^3$ molecule$^{-1}$ s$^{-1}$) of MTs and SQTs
with ozone.

| Species | This study | Atkinson, 1997 | Hoffmann et al., 1997 | IUPAC* (298K) | Pollmann et al., 2005 |
|---|---|---|---|---|---|
| β-Myrcene | 5.93±0.17 | 4.70 | | 4.7 | |
| α-Phellandrene | 28.15±1.11 | 29.8 | | 29 | |
| Limonene | 3.06±0.14 | 2 | 2 | 2.2 | |
| Terpinolene | 19.21±1.20 | 18.8 | | 16 | |
| α-Longipinene | 20.17±0.98 | | | | |
| α-Gurjunene | 14.56±0.15 | | | | |
| β-Caryophyllene | 115.33±3.83 | 116 | 116 | 120 | 110±5.1 |
| β-Caryophyllene-d2 | 129.14±4.42 | | | | |
| Thujopsene | 8.64±0.16 | | | | |
| α-Humulene | 147.04±3.87 | 117 | | 120 | 140±8.8 |
| β-Chamigrene | 15.14±0.16 | | | | |

*IUPAC Task Group on Atmospheric Chemical Kinetic Data Evaluation (http://iupac.pole-ether.fr)