# Peer review of "Calibrating adsorptive and reactive losses of monoterpenes and sesquiterpenes in"

_Atmospheric Measurement Techniques, 2024_

## Author Comment (AC1)

**Reply to comments from Referee #1**

This study utilized 10 monoterpenes and 10 sesquiterpenes, which are commonly emitted from vegetation, to evaluate the reactive and adsorptive losses of volatile organic compounds inside the enclosure using dynamic headspace sampling approaches. In addition, two deuterium-labeled compounds were used to assess whether they could serve as internal standards for evaluating reactive and adsorptive losses of terpenoids during in situ field measurements. Overall, this study is well-written and provide valuable insights into for designing field-based volatile measurement systems to miminize adsorptive and reactive losses. I have a few minor comments that should to be addressed in the revision.

1. It is unclear how the deuterium-labeled compounds were quantified using GC-MS. A detailed description of the quantification method should be provided either in the M&M section or in the supplementary document.

**Reply:** The deuterium-labeled compounds, α-pinene-d3 and β-caryophyllene-d2, were quantified using their characteristic m/z 96 for α-pinene-d3 and m/z 206 for β-caryophyllene-d2. Quantification was based on six-point calibration curves as shown in Fig. S12. We have added this description of the quantification method to Text S2 in the Supplemental Information. While the calibration curves for other general monoterpenes and sesquiterpenes are not included here, they can be found in our previous study (Zeng et al., 2022a).

Zeng, J., Zhang, Y., Zhang, H., Song, W., Wu, Z., and Wang, X.: Design and characterization of a semi-open dynamic chamber for measuring biogenic volatile organic compound (BVOC) emissions from plants, Atmos. Meas. Tech., 15, 79-93, https://doi.org/10.5194/amt-15-79-2022, 2022a.

[Figure]

**Figure S12.** Calibration curves for α-pinene-d3 (m/z 96, a) and β-caryophyllene-d2 (m/z 206, b).

2. It is unclear in which lab experiments the ozone scrubbers were used. Since all experiments appears to have used dry air and if the compressed dry air was free of ozone, it is hard to understand why ozone scrubber is needed. Nonetheless, the author needs to clarify this point to avoid misunderstanding.

**Reply:** As mentioned by the reviewer, ozone scrubber was only used in the experiments for the ozone effect, while it was not used in any other experiments. We had clarified this in Text S1 "*Ozone scrubbers (Zeng et al., 2022a) were used only during the experiment of ozone effect*"

3. While it sounds reasonable to use deuterium-labeled compounds, which are not emitted by the target research subjects, as internal standard in field measurement campaigns to account for the potential adsorptive and reactive losses of plant-derived volatiles, correction factors derived from these deuterium-labeled compounds still should to be explained with caution. This is because adsorptive losses of these deuterium-labeled compounds cannot only occur on the inner surface of the enclosure, but also on the plant surface (especially leaf surface for broad-leaf species) either by passive deposition or active uptake by plants. When the relative contribution of these two process is not known, the applicability of deuterium-labeled compounds for assessing adsorptive losses may be limited. Anyway, I suggest that the authors should address these considerations in the discussion.

**Reply:** Thank you very much for the insightful comments regarding the use of deuterated surrogates as internal standards in field measurements. We completely agree that adsorptive and reactive losses of these compounds may occur not only on the inner surfaces of the chamber but also on plant surfaces (e.g., via passive deposition or active uptake by leaves). We have addressed this in the revised manuscript, specifically in the "Conclusions" section: "*It is important to note that, in addition to adsorptive losses on chamber walls, deuterated surrogates may also be adsorbed on plant surfaces (especially leaf surfaces of broad-leaf species) through passive deposition or active uptake. The relative contributions of wall and plant surface losses to the adsorptive loss are not always known, which may limit the applicability of deuterium-labeled compounds for assessing adsorptive losses. However, a larger wall-to-plant surface area ratio and shorter residence times in the chamber (Zeng et al., 2022a) would make the surrogate method more applicable.*" (L319-324)

4. It is unclear how many replicates were used for each experiment, and what the error bars in the figure or the text refer to (standard deviation or standard error?).

**Reply:** We preformed three replicates for each experimental setting, and the error bars in the figures represent the standard deviation. This has been clarified in the main text (L113-115) "*For each experimental setting, three replicates were performed, and the recovery for a single compound is shown as the mean of these replicates, with error bars representing the standard deviations.*"

Some grammatical errors should be corrected, e.g.,

Line 65, change "existed" to "exists"

**Reply:** Thanks for your carful check. We have revised it in the revised manuscript. (L65)

Line 72 change "by using surrogates like aromatic compounds with quite different reactivity with O3." to "by using surrogates like aromatic compounds that have quite different reactivity with O3."

**Reply:** Following your suggestion, we have got this revised in the main text. (L79)

---

## Author Comment (AC2)

**Reply to comments from Referee #2**

This study evaluates deuterated surrogates (α-pinene-d3 and β-caryophyllene-d2) for tracing adsorptive and reactive losses of monoterpenes (MTs) and sesquiterpenes (SQTs) in dynamic chambers. Using standard gas mixtures, it examines the effects of concentration, temperature, humidity, and O3. Adsorptive losses varied with concentration and temperature, while O3-induced losses were species-specific. The surrogates effectively corrected losses, but additional deuterated compounds are recommended for broader coverage. A strong correlation between adsorption and O3 reactivity underscores the need for careful loss correction, particularly for highly reactive SQTs, which should be measured with O3-free air when correction is unavailable. Overall, the manuscript is well written, and the results are valuable to the literature. I have some minor comments:

Lines 82-83: Please briefly mention the size and dimensions of the chamber, as they affect VOC losses via surface adsorption.

**Reply:** Following your suggestion, we have included the chamber's dimensions in the revised manuscript "*It has a diameter of 25 cm and a height of 28 cm, yielding a volume of 13.7 L.*" (L90)

Section 3.1: Is there an adsorption model that can be used to quantify VOC loss? The authors may refer to He et al. (AMT, 2024) for relevant methods. (Reference: He, L., Liu, W., Li, Y., Wang, J., Kuwata, M., and Liu, Y.: Wall loss of semi-volatile organic compounds in a Teflon bag chamber for the temperature range of 262–298 K: mechanistic insight on temperature dependence, Atmos. Meas. Tech., 17, 755–764, https://doi.org/10.5194/amt-17-755-2024, 2024).

**Reply:** Thank you very much for providing the reference of He et al. (2024) on adsorption modelling. Indeed, previous studies, such as Ghirardo et al. (2020), have proposed whole chamber system calibration to correct for VOC losses, and we also proposed a correction method based on laboratory evaluations (Zeng et al., 2022a). While these methods are useful in ideal laboratory conditions, they seem impractical for field measurements due to: 1) differences in chamber materials and designs; 2) variations in plant leaves and their surfaces; and 3) fluctuating environmental parameters like temperature, light, humidity, and ozone concentrations. Therefore, instead of relying on modelling, we introduce deuterated surrogates to avoid these complexities. In the revised manuscript, we have clarified this in the "Introduction" section with the following addition: "*While previous laboratory studies have evaluated adsorptive losses in empty chambers (Kolari et al., 2012; Hohaus et al., 2016; Helin et al., 2020; Ghirardo et al., 2020; Zeng et al., 2022a; He et al., 2024) and proposed correction methods for simplified chamber systems (Ghirardo et al., 2020; Zeng et al., 2022a; He et al., 2024), variations in chamber design, plant species and environmental conditions introduce additional complexities for field measurements. To address these challenges, internal surrogates are recommended for evaluating and correcting such losses in dynamic chambers (Ortega et al., 2008; Zeng et al., 2022a).*" (L 68-74)

Related references:

Ortega, J., Helmig, D., Daly, R. W., Tanner, D. M., Guenther, A. B., and Herrick, J. D.: Approaches for quantifying reactive and low-volatility biogenic organic compound emissions by vegetation enclosure techniques - Part B: Applications, Chemosphere, 72, 365-380, https://doi.org/10.1016/j.chemosphere.2008.02.054, 2008.

Kolari, P., Back, J., Taipale, R., Ruuskanen, T. M., Kajos, M. K., Rinne, J., Kulmala, M., and Hari, P.: Evaluation of accuracy in measurements of VOC emissions with dynamic chamber system, Atmos. Environ., 62, 344-351, https://doi.org/10.1016/j.atmosenv.2012.08.054, 2012.

Hohaus, T., Kuhn, U., Andres, S., Kaminski, M., Rohrer, F., Tillmann, R., Wahner, A., Wegener, R., Yu, Z., and Kiendler-Scharr, A.: A new plant chamber facility, PLUS, coupled to the atmosphere simulation chamber SAPHIR, Atmos. Meas. Tech., 9, 1247-1259, https://doi.org/10.5194/amt-9-1247-2016, 2016.

Ghirardo, A., Lindstein, F., Koch, K., Buegger, F., Schloter, M., Albert, A., Michelsen, A., Winkler, J. B., Schnitzler, J. P., and Rinnan, R.: Origin of volatile organic compound emissions from subarctic tundra under global warming, Glob. Chang. Biol., 26, 1908-1925, https://doi.org/10.1111/gcb.14935, 2020.

Helin, A., Hakola, H., and Hellén, H.: Optimisation of a thermal desorption–gas chromatography–mass spectrometry method for the analysis of monoterpenes, sesquiterpenes and diterpenes, Atmos. Meas. Tech., 13, 3543-3560, https://doi.org/10.5194/amt-13-3543-2020, 2020.

Zeng, J., Zhang, Y., Zhang, H., Song, W., Wu, Z., and Wang, X.: Design and characterization of a semi-open dynamic chamber for measuring biogenic volatile organic compound (BVOC) emissions from plants, Atmos. Meas. Tech., 15, 79-93, https://doi.org/10.5194/amt-15-79-2022, 2022a.

He, L., Liu, W., Li, Y., Wang, J., Kuwata, M., and Liu, Y.: Wall loss of semi-volatile organic compounds in a Teflon bag chamber for the temperature range of 262-298 K: mechanistic insight on temperature dependence, Atmos. Meas. Tech., 17, 755-764, 10.5194/amt-17-755-2024, 2024.

Lines 136-137: Please add references to this sentence.

**Reply:** We have added two related references in the revised manuscript (L146).

Ortega, J., Helmig, D., Daly, R. W., Tanner, D. M., Guenther, A. B., and Herrick, J. D.: Approaches for quantifying reactive and low-volatility biogenic organic compound emissions by vegetation enclosure techniques - Part B: Applications, Chemosphere, 72, 365-380, https://doi.org/10.1016/j.chemosphere.2008.02.054, 2008.

Niinemets, U., Kuhn, U., Harley, P. C., Staudt, M., Arneth, A., Cescatti, A., Ciccioli, P., Copolovici, L., Geron, C., Guenther, A., Kesselmeier, J., Lerdau, M. T., Monson, R. K., and Peñuelas, J.: Estimations of isoprenoid emission capacity from enclosure studies: measurements, data processing, quality and standardized measurement protocols, Biogeosciences, 8, 2209-2246, https://doi.org/10.5194/bg-8-2209-2011, 2011.

Lines 142-148: Why do reactive VOCs exhibit greater adsorption loss? Any explanation?

**Reply:** Based on our observations, monoterpenes or sesquiterpenes that are more reactive with ozone tend to exhibit greater adsorption losses. While we cannot definitely explain this phenomenon, it may be related to the molecular structure of these compounds. Terpenes, with one or more unsaturated bonds and varying structural configurations (cyclic, bicyclic, or acyclic), show

enhanced reactivity with ozone due to their higher π-electron cloud density. This increased electron density may also facilitate stronger van der Waals interactions with non-polar surfaces, such as chamber walls, thereby increasing their adsorption capacity. We believe it is an interesting observation that warrants further exploration and may inspire future studies. Therefore, we just report our observation results although we do not have a solid explanation at present.

Line 181: Why not use α-pinene or β-caryophyllene directly instead of their isotopes for calibrating losses?

**Reply:** α-Pinene and β-caryophyllene are both monoterpene and sesquiterpene emitted from plants and thus cannot serve as internal surrogates for calibrating losses, as they are directly relevant to plant emissions. Internal surrogates need to meet two criterions: 1) they should not be present in the plant emissions profile or the circulating air, and 2) they should exhibit similar adsorptive behaviors as the target compounds under varying environmental conditions. By using deuterated surrogates, such as α-pinene-d3 and β-caryophyllene-d2, which are plant-irrelevant, we can better mimic the behavior of plant-derived volatiles while avoiding the complexities associated with plant-emitted compounds.

Lines 191-193: Any suggestions for green leaf volatiles such as aldehydes and alcohols? How does humidity impact their losses? Please discuss.

**Reply:** Green leaf volatiles, including aldehydes and alcohols, are indeed emitted by plants and are water soluble, meaning their adsorption may be influenced more by humidity. The impact of humidity on adsorptive loss is complex, as it involves competition for adsorption sites between water molecules and volatiles, as well as potential changes in the energy spectrum of the adsorption sites due to the presence of condensed water. In our previous study (Zeng et al., 2022a), we have evaluated the effects of humidity (RH=0-100%) on the adsorptive loss of some water-soluble compounds, such as acetonitrile, acrylonitrile, acrolein, and acetone, in dynamic chambers. While the humidity effects varied slightly across compounds, we found that the absolute effects were negligible in dynamic chambers. This aligns with findings from other studies (Kolari et al., 2012; Hohaus et al., 2016). Since this study attempted to use deuterated surrogates to calibrate more important monoterpenes and sesquiterpenes, we did not extend our discussion to oxygenated volatiles.

Kolari, P., Back, J., Taipale, R., Ruuskanen, T. M., Kajos, M. K., Rinne, J., Kulmala, M., and Hari, P.: Evaluation of accuracy in measurements of VOC emissions with dynamic chamber system, Atmos. Environ., 62, 344-351, https://doi.org/10.1016/j.atmosenv.2012.08.054, 2012.

Hohaus, T., Kuhn, U., Andres, S., Kaminski, M., Rohrer, F., Tillmann, R., Wahner, A., Wegener, R., Yu, Z., and Kiendler-Scharr, A.: A new plant chamber facility, PLUS, coupled to the atmosphere simulation chamber SAPHIR, Atmos. Meas. Tech., 9, 1247-1259, https://doi.org/10.5194/amt-9-1247-2016, 2016.

Zeng, J., Zhang, Y., Zhang, H., Song, W., Wu, Z., and Wang, X.: Design and characterization of a semi-open dynamic chamber for measuring biogenic volatile organic compound (BVOC) emissions from plants, Atmos. Meas. Tech., 15, 79-93, https://doi.org/10.5194/amt-15-79-2022,

2022a.

Line 197: Were O3 scrubbers or denuders applied before cartridge sampling?

**Reply:** Yes, ozone scrubbers were used to scavenge ozone before cartridge sampling when conducting the experiments for the ozone effect.

Lines 245-249: Sesquiterpene emissions from plants may be more sensitive to O3 stress as reported by the literature. Using O3-free air may not reflect real-world conditions. Why not apply a correction factor for sesquiterpene loss due to O3 instead?

**Reply:** While it is true that sesquiterpene emissions may be more sensitive to $O_3$, calculating the reactive loss of sesquiterpenes due to $O_3$ in a controlled environment is straightforward, as it relies on reaction rate constants and $O_3$ concentrations. However, field measurements introduce significant uncertainties due to: 1) diurnal temperature fluctuations that affect temperature-dependent reaction kinetics; 2) operational complexity of continuous $O_3$ monitoring; 3) large uncertainties in reported rate constants (Table S3); and 4) challenges in distinguishing between stomatal $O_3$ uptake and chemical reactions. These factors make it difficult to accurately quantify reactive losses in the field.

To address these challenges, we used deuterated surrogates, which allow for simultaneous quantification of adsorptive and reactive losses under real-world conditions. Our findings demonstrate that highly reactive sesquiterpenes like caryophyllene and α-humulene are subject to substantial $O_3$-driven degradation (Fig. 5). While long-term $O_3$ fumigation can affect sesquiterpene emissions, short-term $O_3$ scavenging during sampling is less likely to impact plant physiology or sesquiterpene emissions, as suggested by Niinemets et al. (2011) and supported by other studies (Helmig et al., 2007, 2013; Ortega et al., 2008).

Related references:

Helmig, D., Ortega, J., Duhl, T., Tanner, D., Guenther, A., Harley, P., Wiedinmyer, C., Milford, J., and Sakulyanontvittaya, T.: Sesquiterpene emissions from pine trees - Identifications, emission rates and flux estimates for the contiguous United States, Environ. Sci. Technol., 41, 1545-1553, https://doi.org/10.1021/es0618907, 2007.

Helmig, D., Daly, R. W., Milford, J., and Guenther, A.: Seasonal trends of biogenic terpene emissions, Chemosphere, 93, 35-46, https://doi.org/10.1016/j.chemosphere.2013.04.058, 2013.

Ortega, J., Helmig, D., Daly, R. W., Tanner, D. M., Guenther, A. B., and Herrick, J. D.: Approaches for quantifying reactive and low-volatility biogenic organic compound emissions by vegetation enclosure techniques - Part B: Applications, Chemosphere, 72, 365-380, https://doi.org/10.1016/j.chemosphere.2008.02.054, 2008.

Niinemets, U., Kuhn, U., Harley, P. C., Staudt, M., Arneth, A., Cescatti, A., Ciccioli, P., Copolovici, L., Geron, C., Guenther, A., Kesselmeier, J., Lerdau, M. T., Monson, R. K., and Peñuelas, J.: Estimations of isoprenoid emission capacity from enclosure studies: measurements, data processing, quality and standardized measurement protocols, Biogeosciences, 8, 2209-2246, https://doi.org/10.5194/bg-8-2209-2011, 2011.

Table 1: It would be useful to include literature-reported reaction rates for compounds in both groups 1 and 2, which helps readers understand the extent of reaction rate differences.

**Reply:** Following your suggestion, we have included literature-reported reaction rates for all tested monoterpenes and sesquiterpenes in Table S3 in the Supplement.

**Table S3.** Rate constants ($k$, $\times 10^{-16}$ cm$^3$ molecule$^{-1}$ s$^{-1}$) of MTs and SQTs with ozone.

| Compounds | $k$ ($\times 10^{-16}$ cm$^3$ molecule$^{-1}$ s$^{-1}$) |
|---|---|
| α-Pinene-d3 | |
| α-Pinene | 0.866 [2]; 0.866 [3]; 0.96 [5] |
| β-Pinene | 0.15 [2]; 0.15 [3]; 0.19 [5] |
| β-Myrcene | 5.93±0.17 [1]; 4.7 [2] |
| α-Phellandrene | 28.15±1.11 [1]; 29.8 [2]; 29 [5] |
| 3-Carene | 0.37 [2]; 0.371 [3]; 0.49 [5] |
| Limonene | 3.06±0.14 [1]; 2 [2]; 2 [3]; 2.2 [5] |
| 1,8-Cineole | 0.15 [5] |
| γ-Terpinene | 1.4 [2]; 1.6 [5] |
| Terpinolene | 19.21±1.20 [1]; 16 [5] |
| Linalool | 4.3 [3] |
| α-Longipinene | 20.17±0.98 [1]; 2.9 [4] |
| Longicyclene | 0.114 [7] |
| α-Copaene | 1.6 [2]; 2.9 [4]; 1.5 [5] |
| α-Gurjunene | 14.56±0.15 [1] |
| β-Caryophyllene | 115.33±3.83 [1]; 116 [2]; 116 [3]; 110 [4]; 120 [5] |
| β-Caryophyllene-d2 | 129.14±4.42 [1] |
| Thujopsene | 8.64±0.16 [1] |
| Aromadendrene | 65 [4] |
| α-Humulene | 147.04±3.87 [1]; 117 [2]; 140 [4]; 120 [5] |
| Alloaromadendrene | 0.12 [6] |
| β-Chamigrene | 15.14±0.16 [1] |

1. this study; 2. Atkinson, 1997; 3. Hoffmann et al., 1997; 4. Pollmann et al., 2005; 5. IUPAC Task Group on Atmospheric Chemical Kinetic Data Evaluation (http://iupac.pole-ether.fr); 6. Yee et al., 2016; 7. Frazier et al., 2022

Lines 273-274: Again, would chamber-specific correction factors for different VOCs be more effective?

**Reply:** As stated above, chamber- and compound-specific correction factors are typical more applicable to simple, empty chamber systems (Zeng et al., 2022a; He et al., 2024). However, in the field, the complex and variable nature of plant surfaces, physiological processes, and environmental conditions makes such correction factors impractical.

Zeng, J., Zhang, Y., Zhang, H., Song, W., Wu, Z., and Wang, X.: Design and characterization of a semi-open dynamic chamber for measuring biogenic volatile organic compound (BVOC) emissions from plants, Atmos. Meas. Tech., 15, 79-93, https://doi.org/10.5194/amt-15-79-2022,

2022a.

He, L., Liu, W., Li, Y., Wang, J., Kuwata, M., and Liu, Y.: Wall loss of semi-volatile organic compounds in a Teflon bag chamber for the temperature range of 262-298 K: mechanistic insight on temperature dependence, Atmos. Meas. Tech., 17, 755-764, 10.5194/amt-17-755-2024, 2024.